# Visualising historical changes in air pollution with the Air Quality Stripes

Kirsty J. Pringle[1,2], Richard Rigby[3], Steven T. Turnock[4,5], Carly L. Reddington[3],
Meruyert Shayakhmetova[1], Malcolm Illingworth[1], Denis Barclay[1,2], Neil Chue Hong[1,2], Ed Hawkins[6],
Douglas S. Hamilton[7], Ethan Brain[7], and James B. McQuaid[3]

[1]EPCC, University of Edinburgh, UK
[2]Software Sustainability Institute, University of Edinburgh, UK
[3]School of Earth and Environment, University of Leeds, UK
[4]Met Office Hadley Centre, Exeter, UK
[5]University of Leeds Met Office Strategic (LUMOS) Research Group, University of Leeds, UK
[6]National Centre for Atmospheric Science, Department of Meteorology, University of Reading, Reading, UK
[7]Marine, Earth, and Atmospheric Sciences, North Carolina State University, Raleigh, USA

**Correspondence:** Kirsty J. Pringle (k.pringle@epcc.ed.ac.uk), James B. McQuaid (J.B.McQuaid@leeds.ac.uk)

**Abstract.** This paper introduces the Air Quality Stripes, a data visualisation project which presents historical changes in outdoor particulate matter air pollution ($PM_{2.5}$) concentrations across major cities worldwide. Inspired by the popular Warming Stripes image showing trends in surface temperature, the Air Quality Stripes aim to make complex information about air quality trends understandable and engaging for a broad audience. A historical $PM_{2.5}$ dataset (1850–2022) was created by integrating satellite observations with model simulations (with a bias correction step to ensure a smooth time series and address known model biases). Images were produced in collaboration with a visual design specialist and revised after informal feedback from potential audiences. The images show that trends in $PM_{2.5}$ are varied across the globe; recently there have been significant improvements in air quality in much of Europe and North America but worsening air quality in parts of Asia, Africa and South America. By showcasing historical data in easy to interpret images, the project aims to inspire dialogue among individuals, communities, and policymakers about proactive strategies to combat air pollution.

## 1 Introduction

Air pollution poses a major public health risk, contributing to 4.7 million premature deaths in 2021, 89% of which occurred in low- and middle-income countries (Institute for Health Metrics and Evaluation, 2021; Brauer et al., 2024; World Health Organization, 2021a). Despite strong evidence of its health impacts, improving public understanding is vital to drive policy changes, individual actions, and address the social and economic inequalities linked to poor air quality. The Air Quality Stripes aims to raise public awareness and understanding of outdoor air pollution. The images show the change in air quality ($PM_{2.5}$) through time, from the Industrial Revolution to near present day, in cities around the world. We focus on the air pollutant $PM_{2.5}$ because its health impact surpasses that of other pollutants due to its ability to penetrate vital organs and disrupt physiological processes (Schraufnagel et al., 2019b).

The project was inspired by the hugely popular Warming Stripes project (https://showyourstripes.info/), led by Prof Ed Hawkins, which shows the net change in annual mean global mean temperatures as a distinctive series of blue, white and red stripes (Hawkins et al., 2024, submitted). The Warming Stripes have very successfully translated complex scientific data into an easily digestible format that resonates with a wide audience (e.g. The Guardian 2021; BBC Future 2023). The success of the images has inspired other projects including the Biodiversity Stripes (https://biodiversitystripes.info/global, Richardson (2023)) and the Ocean Acidification Stripes (https://oceanacidificationstripes.info/) .

## 1.1  Particulate matter air pollution

Air pollution encompasses various harmful substances (e.g. $NO_2$, $O_3$, metals, $SO_2$), but this study focuses on fine particulate matter ($PM_{2.5}$), composed of airborne particles smaller than 2.5 micrometers. Exposure to $PM_{2.5}$ is strongly linked to adverse health effects, including asthma, lung cancer, and heart disease (Schraufnagel et al., 2019a; World Health Organization, 2021c; Wan Mahiyuddin et al., 2023). Over 99% of the world's population resides in areas where $PM_{2.5}$ levels exceed the World Health Organization's (WHO) air quality guideline value of 5 $\mu$g m$^{-3}$ (World Health Organization, 2021a), but concentrations vary significantly around the globe.

The concentration of PM2.5 at any given location is governed by the balance between the magnitude of source terms (including direct emissions, secondary formation, and transport from other regions) and removal processes (such as deposition and atmospheric dispersion). Sources of PM2.5 are diverse and include combustion from vehicles, residential cooking and heating, industry, power generation, and agriculture, as well as secondary aerosol formation from gaseous precursors like SO, NO, and VOCs. Natural and semi-natural sources -such as mineral dust, sea salt, volcanic emissions, and black carbon from wildfires - can also contribute substantially to PM2.5 levels (Reddington et al., 2014; Zhang et al., 2016; Graham et al., 2021; Pai et al., 2022). Meteorological conditions strongly modulate both source and removal terms. Temperature inversions, for instance, can suppress vertical mixing and trap pollutants near the surface, resulting in acute pollution episodes. Cities situated in valleys or at the foot of mountain ranges can experience higher concentrations due to restricted air movement (limiting the dispersion of pollutants), while some coastal cities are influenced by strong onshore winds that transport pollutants away (Pillai et al., 2002; Dall'Osto et al., 2010). Furthermore, PM2.5 also displays quite significant seasonal trends due to seasonal changes in meteorology and emission patterns. $PM_{2.5}$ has an atmospheric lifetime of approximately a few weeks, allowing pollution to be transported to nearby cities and countries, causing trans-boundary air pollution issues (Zhang et al., 2017; Chen et al., 2022). At the same time, it means that $PM_{2.5}$ concentrations can respond Relatively quickly - on the order of days to weeks - to effective emissions controls and clean air legislation (e.g., Silver et al. 2020 ), making policy interventions both impactful and measurable over short timescales.

## 2 About the data

### 2.1 Combined model and satellite data

Satellite observations and computer model simulations were combined to generate a historical $PM_{2.5}$ time series from 1850 to 2022. Post-2000 data come from a publicly available product integrating satellite and ground-based observations into a gridded global dataset at $0.1°$ resolution (Van Donkelaar et al., 2021). For pre-2000 years, historical trends rely on Earth System Model simulations (Turnock et al., 2020; Sellar et al., 2020). These simulations are part of the Coupled Model Intercomparison Project (CMIP6), where over 30 models provided historical climate simulations for 1850–2014 (Eyring et al., 2016), supporting the Intergovernmental Panel on Climate Change (Arias et al., 2021). All CMIP6 models use the Community Emissions Data System (CEDS) for anthropogenic and wildfire emissions (Hoesly et al., 2018; van Marle et al., 2017), while natural emissions (dust, sea spray) are model-calculated and vary between models. Only some CMIP6 models include interactive particulate matter and chemical processes; we used $PM_{2.5}$ data from one of these models UKESM1 (UK Earth System Model, version 1, Sellar et al. 2020). This model uses a two-moment aerosol microphysics scheme for the main types of particulate matter (sulfate, black carbon, organic carbon, sea salt) and a sectional (bin) scheme for mineral dust (Mulcahy et al., 2020). At present, UKESM1 does not treat nitrate aerosol. Model simulations were at N96 spatial resolution ($1.875°$ longitude $\times 1.25°$ latitude). This model has been widely used for air quality studies (Reddington et al., 2023; Turnock et al., 2020, 2023; Allen et al., 2021; Butt et al., 2017). CMIP6 data are freely available (Earth System Grid Federation, 2024).

Modelling global air pollutant concentrations is challenging, and models are continually refined through comparisons with observational data to improve the accuracy of physical and chemical process representations. Previous studies have shown that CMIP6 multimodel simulations tend to underestimate $PM_{2.5}$ concentrations relative to observations (Turnock et al., 2020). To address this issue and ensure continuity between model output and satellite data, a bias correction approach is applied. Specifically, for each city, a three-year mean (2000–2002) of satellite-derived $PM_{2.5}$ is compared to a three-year mean of modelled concentrations for the same period. The ratio between these means is used to adjust the model values throughout the historical simulation period (1850–2000), correcting for bias. This method is in line with approaches used in previous research (Turnock et al., 2023; Reddington et al., 2023). More information on bias correction techniques is available in Staehle et al. (2024). The average bias correction factor for all cities is 2.65 (St Dev = 1.6), indicating a general underestimate by the model compared to satellite-derived observations. There was found to be low sensitivity to the averaging period chosen (e.g. 10 year average gave a overall bias correction factor of 2.7). Plots showing the model values, satellite-derived values, and bias for each city are publicly available (Pringle, 2024). Due to the scarcity of historical $PM_{2.5}$ observations, evaluating the accuracy of this approximation is challenging. Observations of PM2.5 did not become routine until after the 1990s, prior to that some measurements of "soot" exist, which can give some indirect indication of PM2.5 concentrations, but even these observations are sparse and do not have good temporal or geographical coverage. The concentration data pre 1998 is therefore unconstrained by measurements. However, this method ensures that the historical trend in $PM_{2.5}$ is guided by model simulations (informed by the emissions inventory), while absolute values are aligned with more recent satellite-derived observations with global coverage and higher resolution.

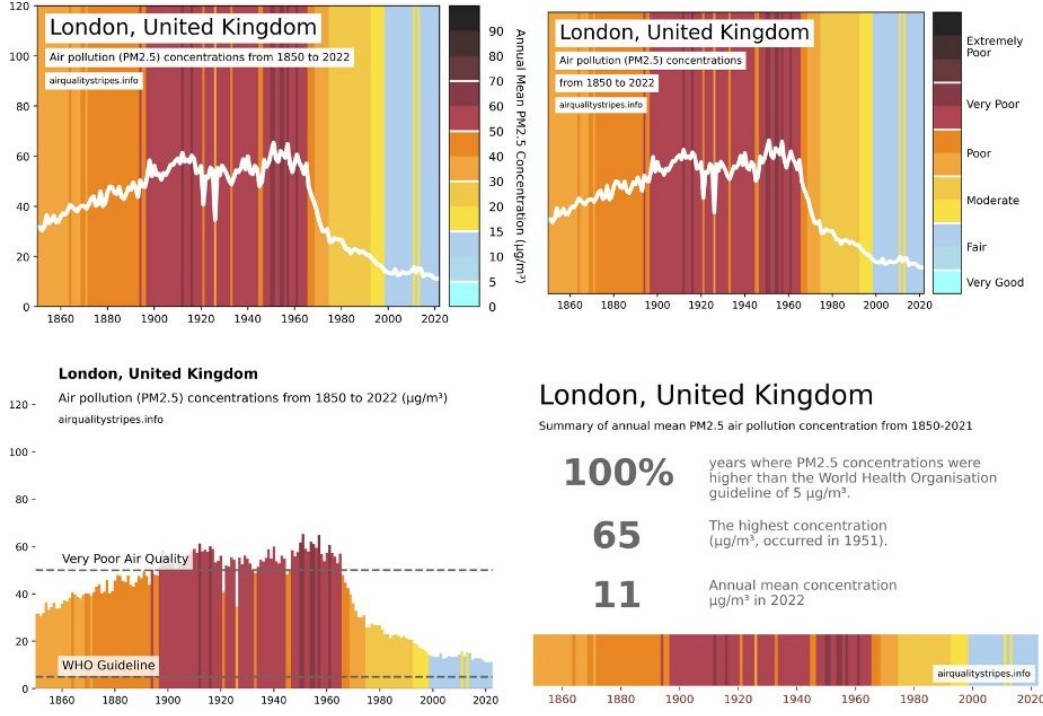

**Figure 1.** The four styles of image created for each city using London (UK) as an example: a) annual mean with absolute values b) annual mean PM$_{2.5}$ concentrations with indicative descriptions, c) annual mean bar plot and d) summary statistics. Images for the other cities are available on the project website: https://airqualitystripes.info/.

## 3 Images and Interpretation

Urban concentrations of PM$_{2.5}$ are most relevant to public health as a large fraction of the world's population live in cities, so for the visualisations, PM$_{2.5}$ concentrations were extracted at the location of major cities around the world. An initial set of images for around 170 cities in over 100 countries was created. To allow for different preferences, four different styles of image were created for each city. Figure 1 shows the four images for a single example city (London, UK).

### 3.1 The Colour of Pollution

The Warming Stripes benefit from an intuitive colour scale (blue is cool, red is warm), which reduces the cognitive load for viewers and makes the plots easy to understand. For air pollution, creating such an intuitive colour palette is more challenging. To bridge the gap between colour and its cognitive associations, it was therefore essential to take into account some of the principles of colour theory. A key aspect of colour theory is understanding the psychological associations between a specific colour, its wider representation in a sociocultural environment, and its representation within the spread of digital media (Christiansen, 2022). For air pollution, creating a simple to understand colour palette is a challenging task due to the often abstract

connections between the term "air pollution" and any specific colour. To address this, multimedia designer Ethan Brain (co-author) analysed colour themes from two hundred images collected from a Google image search for "air pollution". By taking web visuals with the tag "air pollution" and using these images as datasets, visual patterns in popular representation could be formally analysed. As one might expect, polluted images were dominated by reds, browns and greys and clean images by blues. The images were then analysed using a custom sorting and colour grouping tool to identify the dominant colour theme. Then, to hone in on images that best represented air pollution, the initial set of images was filtered only to include results that matched the dominant colours, eliminating outliers and ensuring consistency across the dataset. Dominant colour themes were then extracted from this filtered set and the resulting colours were hand-selected and optimized for web and print colour spaces. Finally, the colours were organized to create a palette that represents increasing values. In the resulting palette the lightest blue represents the cleanest value of concentrations of less than 5 $\mu$g m$^{-3}$. Cities with years in this colour meet the World Health Organisation Air Quality Guideline (World Health Organization, 2021b), all other colours show an exceedance of the guideline value.

## 3.2 Refining the images

The Warming Stripes image notably presents the data without the axes and labels typically associated with scientific data representations, instead opting for a clear, compelling visual that bridges art and science. The Air Quality Stripes images differ in one key way: while the Warming Stripes (and the Biodiversity Stripes) primarily aim to highlight a single, consistent global trend - warming temperatures or biodiversity loss - the changes in PM$_{2.5}$ concentration are more complex and spatially variable. Some regions experience increased concentrations, while others see reductions. It would be challenging to create a single overall image without obscuring these important trends.

However, the project aimed to create accessible and engaging images for a wide audience. To achieve this, different versions of draft images were shown to 15 to 20 individuals, including colleagues (air quality researchers, public engagement specialists and researchers in different fields), friends, and family, during an informal testing phase. Feedback was gathered on aspects such as clarity, visual appeal, colour interpretation, and perceived message. Images with and without 1) a trend line, 2) axes labels, and 3) colour bar labels were tested. The trend line was found to be useful in guiding the eye and reducing the effort required to understand the data. Given the varied regional trends, axes and colour bar labels were deemed essential for clarity. Some individuals expressed uncertainty about interpreting the data, asking questions like, "If concentrations are 10 $\mu$g m$^{-3}$, what does that mean? Is it good or bad?". To address this, plots were updated with indicative labels ("Very Good", "Fair", "Moderate", "Poor", "Very Poor" and "Extremely Poor"). While similar labels are commonly used (e.g., European Environment Agency 2023), they typically reference daily, not annual, mean values. Instead, the WHO guideline of 5 $\mu$g m$^{-3}$ was used for the "Very Good" category, and other categories were estimated by extrapolating from daily indicative levels. To enable direct visual comparison the mapping from PM$_{2.5}$ concentration to the indicative label is the same for every city, for more details see Pringle (2025). White breakpoints were added to the colour bar to highlight the separation between these categories. Finally, a landing page was added to the website with images from four cities, annotated to highlight events that affected PM$_{2.5}$ concentrations

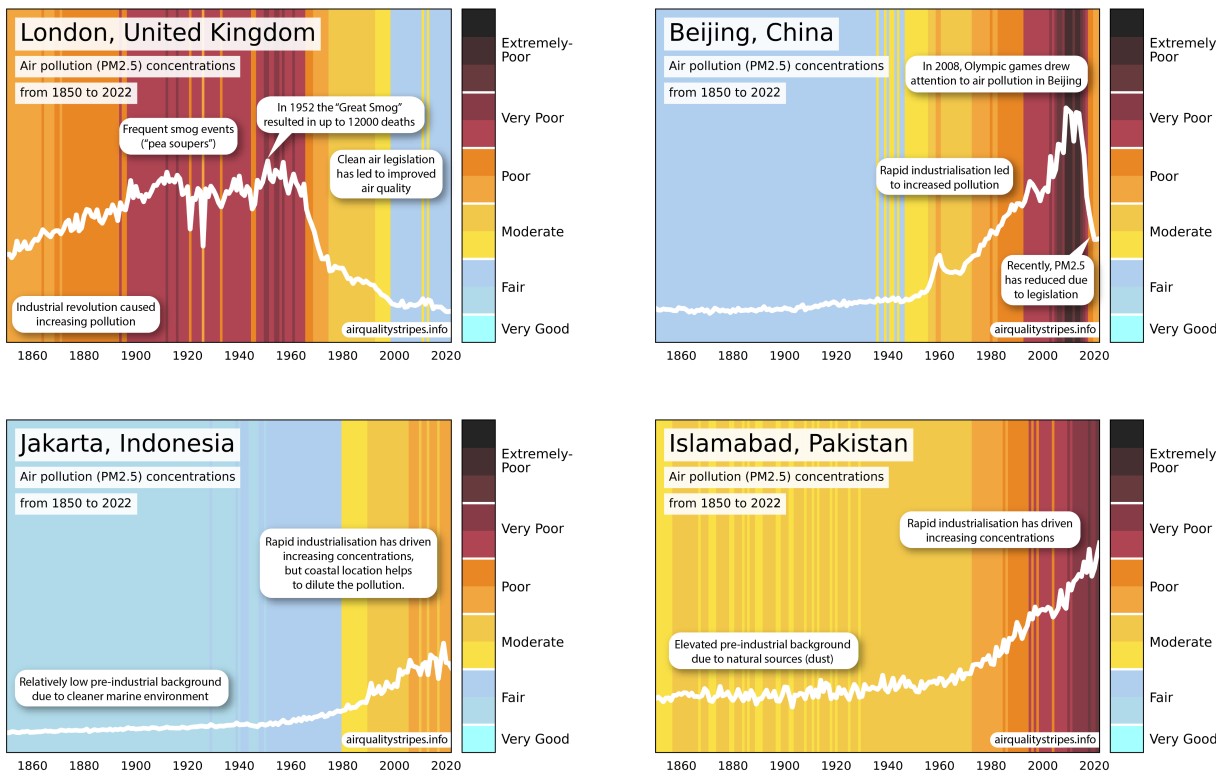

**Figure 2.** Example images for different cities annotated to highlight causes of the change in PM2.5 which are used on the landing page of the project website.

## 4 Discussion and Conclusion

### 4.1 Air Pollution Trends

The images show that $PM_{2.5}$ levels have reduced in many cities. Concentrations in much of Europe and North America have been declining since the 1960s, and very rapid reductions have been achieved in China in the past decade. This offers real hope for public health; $PM_{2.5}$ concentrations can respond rapidly to effective air quality legislation, and many cities have made great strides in reducing these levels. London has lower $PM_{2.5}$ concentrations today than it has had at any point in the past 150 years! Unfortunately, many cities in Central and southern Asia face worsening air pollution. There is an urgent need to address these increasing $PM_{2.5}$ concentrations as huge numbers of people experience high concentrations on a daily basis. The images also show the dominance of natural of $PM_{2.5}$. Regions with high dust and wildfire emissions (e.g. India, Pakistan and much of northern Africa) may struggle to achieve the WHO recommendations, even with very stringent emission controls (Pai et al., 2022).

It is important to note that these images do not show what is driving the changes in air quality. While some of the progress seen in the Global North has been achieved through local measures such as clean air zones, reductions in coal burning, and stricter air quality legislation, other improvements have occurred through the outsourcing of heavily polluting industries to the Global South. This dynamic has shifted some of the pollution burden from wealthier to poorer regions, contributing to persistent or worsening air quality and health issues in parts of the Global South (Nansai et al., 2020). This disparity underscores environmental and social justice concerns, as communities in lower-income regions disproportionately bear the health and environmental impacts of industrial pollution generated for the benefit of wealthier economies.

## 4.2 Learnings from Development of the Images

Through the process of creating and refining these images, several valuable insights were gained:

1. **Collaboration with Visual Experts**: Partnering with visual design specialists helped in the development of compelling and accessible graphics. Their expertise in colour theory helped create a fnal product that was visually appealing and intuitive.

2. **Informal Feedback and Review Stage:** Feedback from colleagues, friends, and family was invaluable. While this project used informal connections, many visualisation efforts could benefit from a structured feedback process to ensure accessibility and clarity for diverse audiences. Any further development of these images or associated works, will involve a formal structured review process which will take feedback from a diverse range of stakeholders

3. **City-Specific Focus**: Creating data for individual cities made the images relatable and grounded. This approach allows the viewer to connect with the information on a more personal level.

4. **Selected Annotations**: Narrative annotations on a subset of images made the data more relatable, providing context and highlighting significant points. They also helped viewers better understand the overall structure of the images.

The Air Quality Stripes project received significant media attention (Fuller, 2024; Hunter, 2024; Turns, 2024; McQuaid et al., 2024), highlighting the power of visual data to engage and inform the public. Future efforts will aim to deepen this engagement by using these images to facilitate discussions about lived experiences with air pollution worldwide, through methods such as workshops, storytelling, or other participatory approaches.

*Code and data availability.* The data used to create the Air Quality Stripes is available (Pringle and McQuaid, 2024) as is the visualisation code used (Pringle and Rigby, 2024).

*Author contributions.* JM and KP led the project with scientific guidance from CR and ST, software support from RR and MI and visualisation guidance from DH, EB, EH, MS and DB.

*Competing interests.* At least one of the authors is a member of the editorial board of Geoscience Communication.

*Ethical Statement* This article does not include any studies involving human or animal subjects. The air pollution stripes build
on a previously established tool—the global warming stripes—and were therefore evaluated informally and anonymously, as
clearly stated in the paper. Due to this, ethical review was therefore not required.

*Acknowledgements.* KP, DB and NCH were supported by UKRI grants EP/S021779/1 and AH/Z000114/1 for the Software Sustainability
Institute. This work (JM) was supported by the Natural Environment Research Council [NE/T010401/1]. KP and MI are grateful for two
internal grants from EPCC, University of Edinburgh. EH is supported by the National Centre for Atmospheric Science and the Ireland-UK
Co-Centre for Climate + Biodiversity + Water. MS was supported through Access to Industry's Access to Data program. The contributions of
ST were funded by the Met Office Climate Science for Service Partnership (CSSP) China project under the International Science Partnerships
Fund (ISPF). This work used JASMIN, the UK's collaborative data analysis environment (https://www.jasmin.ac.uk, Lawrence et al. (2013)).

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
