# Peer review of "Visualising historical changes in air pollution with the Air Quality Stripes"

_EGUsphere, 2024_

## Referee Comment (RC3)

This manuscript presents a data visualisation method designed to help users quickly grasp trends in air pollution and understand its impacts on human health over several decades. Communicating air quality levels, including their causes and implications for health, is often challenging. This tool has the potential to simplify and enhance such communication. The final graphs are certainly helpful, as they integrate a lot of information in a clear and understandable manner. It could be especially valuable for policymakers, NGOs, and the general public to raise awareness and support informed decision-making. Using WHO Guidelines as a benchmark is a strong point and provides a credible standard for comparison.

However, I have some concerns and suggestions that the authors may wish to consider to significantly enhance the manuscript's clarity, robustness, and utility.

**General:** Is the project focused solely on PM2.5? While PM2.5 is a key pollutant, referring more generally to "air quality" without including other pollutants (e.g., $NO_2$, $O_3$) may be misleading. This is particularly important when comparing across European countries, where long-range transport rather than local emissions can heavily influence PM2.5 levels. If the project is to extend beyond PM2.5, what needs to be done to make it easily applicable to other pollutants?

**Specific per section:**

**Section 1.1:**

**L34–35:** How does onshore flow affect PM2.5 levels? Does it lead to an increase due to sea salt or a decrease due to dilution? This should be clarified for the reader who is not so familiar with this topic.

**L37:** The statement about the "short lifetime" of PM2.5 could be expanded. What is the typical lifetime? Why is this relevant? Providing a brief explanation would enhance the reader's understanding.

**Section 2.1:**

While it is understandable that the authors use climate simulations to support their visualisations, it is important to acknowledge that most climate models are not optimised for air quality assessment, particularly in urban contexts. The manuscript notes that many CMIP6 models do not include key physical and chemical processes relevant to PM2.5, which is a significant limitation, especially given the focus on urban environments and long-range transport that affects cities' background concentrations.

The authors are commended for applying bias correction, which is essential in this context. However, more detail on the model's spatial resolution is needed. Climate models generally operate on coarse grids, which may not capture city-level variations in PM2.5.

**L49:** Was UKESM1 selected because it includes relevant particulate matter processes? If so, this should be clearly stated. If not, the rationale for choosing this model should be explained.

**Section 3.1:**

**L75:** Is there a reference or source for the colour theory used in the visualisation? If so, it should be included.

**L97:** How many individuals participated in the informal testing phase, and what kind of feedback was collected? More information on the methodology would strengthen the section.

**L104:** What daily values were used to extrapolate the ranges for the remaining categories? Were these values consistent across all cities? If not, the differences should be clearly indicated, as this affects interpretation.

---

## Author Comment (AC1)

**Reviewer 1**

This is a charming paper with very useful messages. The ability for stakeholders to quickly identify the air pollution situation in different cities is beneficial, as is the ability to see the chronology of air pollution in the cities and how this chronology can rapidly change when successful clean air action is implemented. The paper is novel in its approach and should be published once the following relatively small comments have been addressed.

*We thank the reviewer for their kind comments and for the positive review.*

The paper should acknowledge more fully that there are very few measurements of air pollution anywhere in the world before the 1950s and hence the model outputs are very unconstrained, especially before the satellite period.

*We agree with the reviewer and we have made the following changes to the text in the revised manuscript:*

**Previous version:**

[revised manuscript text omitted]

Line 36 "…atmospheric lifetime of a few weeks,…" should be changed to "…atmospheric lifetime of approximately a few weeks,…"

Changed (please see above).

---

## Author Comment (AC2)

**Reviewer 2**

The manuscript "Visualising historical changes in air pollution with the Air Quality Stripes" describes a data visualization project aimed at making "complex information about air quality trends understandable and engaging for a broad audience". Overall, the manuscript provides an interesting approach that stems from the well-known "Warming Stripes", which should help resonating with different audiences given their recognisability.

The manuscript is well structured, as it presents title, authors, abstract, introduction, information about the data, how the images are developed, discussion and conclusion. References seem overall appropriate. All figures are mentioned and described in the text.

The impact these "Air Quality Stripes" may have on the communication with different stakeholders is evident.

*We thank the reviewer for their review and for these positive comments.*

However, I believe that a "formal testing" phase would benefit the project (and the paper). Although the feedback provided by "colleagues, friends and family" (line 97) certainly provided interesting and constructive suggestions, the involvement of other testers (with different backgrounds and roles) could help making this data visualisation tool even more "accessible and engaging" (line 96) for different audiences, as the authors themselves underline (lines 135-137).

*We agree that a formal testing phase involving a broader range of users would have been beneficial; the potential value of broader, structured feedback is something we acknowledge and recommend as part of future work. We note that a formal testing phase is not commonly included in the development of static images for public engagement, however a formal user testing phase is more common in the development of interactive tools; in future we plan to learn from the testing strategy used by this community.*

*We do not intend to change these original images as they are now well established in the community, but we are working on further developing them (including images of future PM2.5 predictions and animated images), these future developments will be subject to a formal testing phase with a wide range of stakeholders.*

*We have made the following changes to the revised manuscript to add more detail and clarification.*

**Previous Version**

2. Informal Feedback and Review Stage: Feedback from colleagues, friends, and family was invaluable. While this project used informal connections, many visualisation efforts could benefit from a structured feedback process to ensure accessibility and clarity for diverse audiences.

**New Version**

2. Informal Feedback and Review Stage: Feedback from colleagues, friends, and family was invaluable. While this project used informal connections, many visualisation efforts could benefit from a structured feedback process to ensure accessibility and clarity for diverse audiences. **The visualisations have been heavily used e.g. by the EPIC Air Quality Fund community for engagement and outreach, and additional cities have been added on request   Any further development of these images or associated works, will**

**involve a formal structured review process which will take feedback from a diverse range of stakeholders.**

**Previous Version**

However, the project aimed to create accessible and engaging images for a wide audience. To achieve this, draft images were shown to colleagues, friends, and family during an informal testing phase.

**New Version**

However, the project aimed to create accessible and engaging images for a wide audience. **To achieve this, different versions of draft images were shown to approximately 20 individuals, including colleagues (air quality researchers, public engagement specialists and researchers in different fields), friends, and family, during an informal testing phase. Feedback was gathered on aspects such as clarity, visual appeal, colour interpretation, and perceived message.**

Overall, the work presented in the paper can certainly have an impact on air quality communication, considering also the annotated images available also on the website, that guide the users.

---

## Author Comment (AC3)

**Reviewer 3**

This manuscript presents a data visualisation method designed to help users quickly grasp trends in air pollution and understand its impacts on human health over several decades. Communicating air quality levels, including their causes and implications for health, is often challenging. This tool has the potential to simplify and enhance such communication. The final graphs are certainly helpful, as they integrate a lot of information in a clear and understandable manner. It could be especially valuable for policymakers, NGOs, and the general public to raise awareness and support informed decision-making. Using WHO Guidelines as a benchmark is a strong point and provides a credible standard for comparison.

*We thank the reviewer for the thorough review, and these encouraging comments.*

However, I have some concerns and suggestions that the authors may wish to consider to significantly enhance the manuscript's clarity, robustness, and utility.

**General:** Is the project focused solely on PM2.5? While PM2.5 is a key pollutant, referring more generally to "air quality" without including other pollutants (e.g., $NO_2$, $O_3$) may be misleading. This is particularly important when comparing across European countries, where long-range transport rather than local emissions can heavily influence PM2.5 levels. If the project is to extend beyond PM2.5, what needs to be done to make it easily applicable to other pollutants?

*This is an important point. This project focuses on particulate matter (PM2.5) due to its significant health impacts; the Global Burden of Disease study estimates PM2.5 contributes to ~4 million premature deaths a year, compared to 350,000 for Ozone. Similarly the Disability-Adjusted Life Years estimated for $NO_2$ is lower than that for PM2.5 or Ozone. PM2.5 is the pollutant of most concern for air quality and health, thus it is the focus of this first attempt to present global historical air quality data in an engaging way.*

*Due to the very high public health burden, particulate matter air pollution has received significant press attention over the past decade and it is now a fairly well-understood concept and a widely recognised problem, thus images that focus solely on PM2.5 are an effective entry point for broader engagement on air quality issues. NGOs and policy makers will also be aware of the term.*

*To avoid confusion, every image has the words "Air Pollution (PM2.5) Concentrations" on it and we have a section on the website giving a background on PM2.5 (https://airqualitystripes.info/faq/).*

*We agree that it would be interesting to also consider other air pollutants and we plan to do this in the future, but this requires a substantial amount of work, not just to combine present day observations (which are less widely available for $O_3$ and $NO_2$) and historical model data, but also to consider how to present the data. The different pollutants differ substantially in terms of sources, atmospheric behaviour, and trends ($NO_2$ often reflects local emissions and shows sharper spatial and temporal gradients, while $O_3$ has a more complex, non-linear relationship with precursors) and they all have very different historical profiles. This complexity means that simply combining the concentrations of all pollutants in a single image would obscure or "smear out" the clear trends that are visible in PM2.5, making the images less engaging and the message and interpretation more complex. Future work will consider how to address this.*

Added text in Intro:

**We focus on the air pollutant PM2.5 because its health impact surpasses that of other pollutants due to its ability to penetrate vital organs and disrupt physiological processes (Schraufnagel et al., 2019b))**

**Specific per section:**

**Section 1.1: L34–35:** How does onshore flow affect PM2.5 levels? Does it lead to an increase due to sea salt or a decrease due to dilution? This should be clarified for the reader who is not so familiar with this topic.

*Thanks for this suggestion. We have now re-written this section to include more detail and to clarify this point.*

**Previous Version**

PM2.5 concentrations are influenced by factors such as industrial and agricultural activity, urbanization, air quality regulation, geographic location and meteorological conditions. In some places natural (or semi-natural) sources of PM2.5 such as mineral dust or black carbon from wildfires strongly influence concentrations (Reddington et al., 2014; Zhang et al., 2016; Graham et al., 2021; Pai et al., 2022), while other urban locations are affected by strong onshore winds which can influence concentrations (Pillai et al., 2002; Dall'Osto et al., 2010). PM2.5 has an atmospheric lifetime of a few weeks, allowing pollution to travel between nearby cities and countries, causing transboundary air pollution issues (Zhang et al., 2017; Chen et al., 2022). However, this relatively short lifetime means that concentrations respond quickly to effective clean air legislation (e.g. Silver et al. 2020).

**New Version**

**The concentration of PM2.5 at any given location is governed by the balance between the magnitude of source terms (including direct emissions, secondary formation, and transport from other regions) and removal processes (such as deposition and atmospheric dispersion). Sources of PM2.5 are diverse and include combustion from vehicles, residential heating, industry, power generation, and agriculture, as well as secondary aerosol formation from gaseous precursors like $SO_2$, $NO_x$, and VOCs. Natural and semi-natural sources - such as mineral dust, sea salt, volcanic emissions, and black carbon from wildfires - can also contribute substantially to PM2.5 levels (Reddington et al., 2014; Zhang et al., 2016; Graham et al., 2021; Pai et al., 2022).**

**Meteorological conditions strongly modulate both source and removal terms. Temperature inversions, for instance, can suppress vertical mixing and trap pollutants near the surface, resulting in acute pollution episodes. Cities situated in valleys or at the foot of mountain ranges can experience higher concentrations due to restricted air movement (limiting the dispersion of pollutants), while some coastal cities are influenced by strong onshore winds that transport pollutants away (Pillai et al., 2002; Dall'Osto et al., 2010). Furthermore, PM2.5 also displays quite significant seasonal trends due to seasonal changes in meteorology and emission patterns.**

**PM2.5 has an atmospheric lifetime of approximately a few weeks, allowing pollution to be transported to nearby cities and countries, causing transboundary air pollution issues (Zhang et al., 2017; Chen et al., 2022). However, this relatively short lifetime**

**means that concentrations respond quickly to effective clean air legislation (e.g. Silver et al. 2020).**

**L37:** The statement about the "short lifetime" of PM2.5 could be expanded. What is the typical lifetime? Why is this relevant? Providing a brief explanation would enhance the reader's understanding.

*Thanks for this comment. We agree and we have expanded the description of the lifetime of PM2.5 and why it is relevant for air quality studies.*

**Previous Version**

PM2.5 has an atmospheric lifetime of a few weeks, allowing pollution to travel between nearby cities and countries, causing transboundary air pollution issues (Zhang et al., 2017; Chen et al., 2022). However, this relatively short lifetime means that concentrations respond quickly to effective clean air legislation (e.g. Silver et al. 2020).

**New Version**

PM2.5 has an atmospheric lifetime of **approximately** a few weeks, allowing pollution to be transported to nearby cities and countries, causing transboundary air pollution issues (Zhang et al., 2017; Chen et al., 2022). **At the same time, it means that PM2.5 concentrations can respond relatively quickly - on the order of days to weeks - to effective emissions controls and clean air legislation (e.g., Silver et al., 2020), making policy interventions both impactful and measurable over short timescales.**

**Section 2.1:** While it is understandable that the authors use climate simulations to support their visualisations, it is important to acknowledge that most climate models are not optimised for air quality assessment, particularly in urban contexts. The manuscript notes that many CMIP6 models do not include key physical and chemical processes relevant to PM2.5, which is a significant limitation, especially given the focus on urban environments and long-range transport that affects cities' background concentrations.

*Thanks for this comment, we have added more detail on and references to the model (see below).*

The authors are commended for applying bias correction, which is essential in this context. However, more detail on the model's spatial resolution is needed. Climate models generally operate on coarse grids, which may not capture city-level variations in PM2.5.

*The plots show annual mean concentrations of PM2.5 over a historical timescale; using the bias correction approach gives us a higher resolution, observationally constrained baseline for the present day (1998 onwards) we then apply the longer term trends (from the model) to this corrected baseline.  Thus while the model data is on a coarse grid, because the satellite data is at a higher resolution (0.1 degree), the historical concentrations are informed by the higher resolution data.*

Model and satellite resolution has been added to the text

**L49:** Was UKESM1 selected because it includes relevant particulate matter processes? If so, this should be clearly stated. If not, the rationale for choosing this model should be explained.

*Thanks for highlighting this lack of clarity, we have updated the text.*

**Previous version**

Only some CMIP6 models include interactive particulate matter and chemical processes; we used PM2.5 data from UKESM1 (UK Earth System Model, version 1, Sellar (2020)).

**New version**

Only some CMIP6 models include interactive particulate matter and chemical processes; we used PM2.5 data from one of these models, UKESM1 (UK Earth System Model, version 1, Sellar et al. 2020). This model uses a two-moment aerosol microphysics scheme for the main types of particulate matter (sulfate, black carbon, organic carbon, sea salt) and a sectional (bin) scheme for mineral dust (Mulcahy et al., 2020). At present, UKESM does not treat nitrate aerosol. Model simulations were at N96 spatial resolution (1.875° longitude × 1.25° latitude). This model has been widely used for air quality studies (Reddington et al., 2023; Turnock et al., 2020, 2023; Allen et al., 2021; Butt et al., 2017). CMIP6 data are freely available (Earth System Grid Federation, 2024).

Also added resolution of the satellite: "0.1 degree resolution"

**Section 3.1: L75:** Is there a reference or source for the colour theory used in the visualisation? If so, it should be included.

Added reference to: Building Science Graphics: An Illustrated Guide to Communicating Science through Diagrams and Visualizations by Jen Christiansen

**L97:** How many individuals participated in the informal testing phase, and what kind of feedback was collected? More information on the methodology would strengthen the section.

**Previous Version**

However, the project aimed to create accessible and engaging images for a wide audience. To achieve this, draft images were shown to colleagues, friends, and family during an informal testing phase.

**New Version**

However, the project aimed to create accessible and engaging images for a wide audience. **To achieve this, different versions of draft images were shown to 15 to 20 individuals, including colleagues (air quality researchers, public engagement specialists and researchers in different fields), friends, and family, during an informal testing phase. Feedback was gathered on aspects such as clarity, visual appeal, colour interpretation, and perceived message.**

**L104:** What daily values were used to extrapolate the ranges for the remaining categories? Were these values consistent across all cities? If not, the differences should be clearly indicated, as this affects interpretation.

*We've added a description on the mapping values to our Air Quality Stripes Zenodo repository https://zenodo.org/records/15363039. The values were consistent across all cities, so any given PM2.5 concentration always maps to the same indicative category.*

Added: **To enable direct visual comparison the mapping from PM2.5 concentration to the indicative label is the same for every city, for more details see Pringle (2025)**

---

## Author Response (AR3)

1. Ethical issue

The use and usability of stripes like this has been proven by the warming stripes that have previously been published and widely used. While the authors did receive feedback on the development of the air pollution stripes, they were open about this feedback being informal. Since the feedback was also anonymous in this setting, I accept the ethical statement as it stands that no human subjects were used directly. However, I think it would be useful and honest for the readers if the ethical is updated accordingly. Maybe something like:

"This article does not include any studies involving human or animal subjects. The air pollution stripes build on a previously established tool—the global warming stripes—and were therefore evaluated informally and anonymously, as clearly stated in the paper. Due to this, ethical review was therefore not required."

Thanks for the suggestion, we have added this text to the ethical statement.

2. Abstract length

As it states in the guidelines for GC Letters, the Abstract needs to be 200 words. Could you please edit the Abstract so that it is within this limit?

I have double checked the abstract length, it is 177 words.

3. Dataset

I tested the link for the dataset Air Quality Stripes: Comparison of Model and Satellite Values at http://dx.doi.org/10.5281/zenodo.14392693 several times on different browsers and I only received a "504 Gateway Time-out" error message. Please ensure that this dataset is accessible or update the link accordingly.

Apologies, I am not sure what is happening here. I have double checked the links in the References section and they link works for me and my colleagues. I have updated the link in the manuscript assets to be the full URL not just the doi, so these works directly as links too. There is an Air Quality Stripes "community" on Zenodo where all the assets related to this work are stored and open to access.

The Air Quality Stripes images visualise historical changes in particulate matter air pollution in over 150 cities worldwide. The project celebrates significant improvements in air quality in regions like Europe, North America, and China, while highlighting the urgent need for action in areas such as Central Asia. Designed to raise awareness, it aims to inspire discussions about the critical impact of air pollution and the global inequalities it causes.